# Effects of Higher Normal Blood Pressure on Brain Are Detectable before Middle-Age and Differ by Sex

**DOI:** 10.3390/jcm11113127

**Published:** 2022-05-31

**Authors:** Khawlah Alateeq, Erin I. Walsh, Walter P. Abhayaratna, Nicolas Cherbuin

**Affiliations:** 1Centre for Research on Ageing, Health and Wellbeing, College of Health and Medicine, Australian National University, Canberra 2601, Australia; erin.walsh@anu.edu.au (E.I.W.); nicolas.cherbuin@anu.edu.au (N.C.); 2Radiological Sciences, College of Applied Medical Sciences, King Saud University, Riyadh 11451, Saudi Arabia; 3School of Medicine and Psychology, College of Health and Medicine, The Australian National University, Canberra 2601, Australia; walter.p.abhayaratna@act.gov.au

**Keywords:** blood pressure, neuroimaging, sex, age

## Abstract

Background: To quantify the association between blood pressure (BP) across its full range, brain volumes and white matter lesions (WMLs) while investigating the effects of age, sex, body mass index (BMI), and antihypertensive medication. Methods: UK Biobank participants (*n* = 36,260) aged (40–70) years were included and stratified by sex and four age groups (age ≤ 45, 46–55, 56–65 and > 65 years). Multi-level regression analyses were used to assess the association between mean arterial pressure (MAP), systolic BP (SBP), diastolic BP (DBP), and brain volumes segmented using the FreeSufer software (gray matter volume [GMV], white matter volume [WMV], left [LHCV] and right hippocampal volume [RHCV]) and WMLs. Interaction effects between body mass index (BMI), antihypertensive medication and BP in predicting brain volumes and WMLs were also investigated. Results: Every 10 mmHg higher DBP was associated with lower brain volumes (GMV: −0.19%–−0.40%) [SE = 47.7–62.4]; WMV: −0.20–−0.23% [SE = 34.66–53.03]; LHCV: −0.40–−0.59% [SE = 0.44–0.57]; RHCV: −0.17–−0.57% [SE = 0.32–0.95]) across all age groups. A similar pattern was detected in both sexes, although it was weaker in men. Every 10 mmHg higher MAP was associated with larger WMLs across all age groups but peaked >65 years (1.19–1.23% [SE = 0.002]). Both lower BMI and anti-hypertensive medication appeared to afford a protective effect. Conclusion: Higher BP is associated with worse cerebral health across the full BP range from middle adulthood and into old age.

## 1. Introduction

As the population ages, the proportion of individuals aged 65 and over is predicted to increase to 16% of the population by 2050 worldwide, with severe implications for the prevalence of age-related diseases [1]. As there are currently no disease-modifying treatments for major neurodegenerative conditions such as dementia, a consensus is building around the need to improve preventative efforts and decrease exposure to known modifiable risk factors. Importantly, there is strong agreement that cardiovascular disease is a significant risk factor for cerebral lesions [2], cognitive impairment [3], and dementia [4], However, the extent to which stronger preventive efforts should be aimed at maintaining optimal blood pressure (BP) in early adulthood and across the life course, rather than primarily concentrating on detecting and treating hypertension in middle age, is unclear.

Despite the overwhelming evidence linking hypertension to cerebrovascular disease [5,6,7], and dementia [8,9] the time at which increasing BP starts impairing brain structure and function has not been determined. High BP in middle age has been found to be associated with the development of neurofibrillary tangles [10], amyloid angiopathy [11], as well as brain atrophy [5]. These findings suggest that more subtle brain changes may be occurring earlier in adulthood [12]. In addition, the threshold at which increased BP becomes deleterious is the focus of much debate.

Recent research has revealed that in addition to hypertension, elevated BP in the pre-hypertensive, or even the upper normal range, may be harmful to cerebral health. A recent systematic review estimated that every 10 mmHg increase in systolic BP (SBP) above 120 mmHg is associated with a 1.3% decrease in hippocampal (HC) volume [5]. This is of major clinical significance, as HC atrophy is strongly predictive of conversion to dementia [13]. Studies have also reported that SBP greater than 120 mmHg significantly raises the risk of cognitive impairment [14,15]. Furthermore, the detrimental effects of elevated BP on cerebral health emerge earlier in adulthood (19–40 years) than previously assumed [16]. Taken together, these findings suggest that the adverse effects of elevated BP on cerebral health may start developing at much lower levels than the current thresholds for pre-hypertension. However, more research is needed to precisely characterise these effects.

A complicating factor in elucidating the complex relationship between BP and cerebral health is that it appears to differ in men and women [17,18]. This difference may be due, at least in part, to different exposure to health and lifestyle risk factors including body mass index (BMI), cholesterol, smoking, and physical activity, which are known to modulate the effect of BP on the brain, in men and women. It is therefore important to investigate BP effects in each sex, preferably by stratifying analyses, while considering the contribution of major related risk factors.

The aim of this study was to quantify the associations between BP across its full range and brain volumes as indexes of cerebral health in cognitively healthy individuals. Other aims were to investigate how this association is modified by age, sex, and other risk factors, and to determine which BP measures (mean arterial pressure [MAP], SBP, or diastolic BP [DBP]) most strongly contribute to cerebral health.

## 2. Materials and Methods

### 2.1. Study Design and Participants

The United Kingdom Biobank (UKB) study is a large prospective cohort study that surveys more than 500,000 participants aged between 40 and 70 years at baseline who were recruited from the National Health Service central registers [19]. Of those, only participants who had baseline DBP and SBP measurements (*n* = 456,990) and completed a structural magnetic resonance imaging (MRI) scan at the second assessment (*n* = 36,260) were considered for inclusion. After further excluding participants with neurological disorders (*n* = 3275), 36,260 participants were available for analysis (Appendix B).

The UK Biobank study was approved by the North-West Multi-centre Research Ethics Committee (reference number 06/MRE08/65). All participants provided their consent to take part in the study via an electronic signature. This study followed the Strengthening the Reporting of Observational Studies in Epidemiology (STROBE) guidelines [20].

### 2.2. Blood Pressure

Participants’ brachial BP was measured as the average of two measurements using an Omron M4 monitor (OMRON Healthcare Europe, NA, Hoofddorp), from the upper arm in a seated position using a well-fitting cuff following a minimum 5-min rest period. The participants’ MAP and pulse pressure (PP) were computed using the following formula: MAP mmHg = diastolic BP + (1/3 × [SBP–DBP]) and PP mmHg: (SBP–DBP). BP was categorised according to the 2017 American Heart Association guidelines [21]. Hypertension (stage 2) was defined by SBP ≥ 140 mmHg, or DBP ≥ 90 mmHg, or use of antihypertensive medication, Hypertension (stage 1) by SBP 130–139 mmHg or a DBP 80–89 mmHg, or use of antihypertensive medication, elevated by SBP 120–129 mmHg or a DBP < 80 mmHg and not meeting hypertension criteria and normal by SBP < 120 or DBP < 80 mmHg and not meeting hypertension/elevated criteria.

### 2.3. Image Acquisition

The participants underwent MRI scans at one of three imaging centres during the second (2014+) visits using the same scanner (3T Siemens Skyra, running VD13A SP4 using a 32-channel head coil). A 3D magnetization-prepared rapid acquisition gradient echo sequence was performed over a five-minute duration to acquire T1-weighted images in the sagittal orientation (resolution = 1 × 1 × 1 mm; field of view = 208 × 256 × 256 matrix). Further details of the acquisition parameters have been reported elsewhere [19].

### 2.4. Segmentation and Image Analysis

Images were processed and segmented with FreeSurfer (version 6.0.5) [22]. The FreeSurfer pipeline has been extensively described elsewhere [23], but briefly it involves motion correction, transformation to Talairach image space, inhomogeneity normalization, removal of non-brain tissues using hybrid watershed, volumetric segmentation [24,25], cortical surface reconstruction and parcellation, and a quality control check [26]. The full set of image analysis pipeline scripts has been reported elsewhere [26]. Regions of interest (ROIs) considered in this study were total gray matter volume (GMV), total white matter volume (WMV), left (LHCV) and right hippocampus volume (RHCV), and white matter lesions (WMLs) because these regions and WMLs have been reported to be associated with elevated BP in a recent systematic review [5].

### 2.5. Covariates

The covariates considered included age, sex, education, BMI, serum high-density lipoprotein cholesterol (HDL-C), total cholesterol (TC) level, diabetes mellitus diagnosed by a doctor, self-reported smoking status (never, previous, or current), level of physical activity (days per week engaging in a minimum of ten minutes of continuous energetic activity), and alcohol intake.

### 2.6. Statistical Analyses

All statistical analyses were conducted using the R statistical package (Version 3.2) under RStudio (version 1.2.5019). Group differences were tested using *t*-tests for continuous measures and chi-square tests for categorical measures. The distribution of each variable was inspected, and log transformations were utilized to transform skewed data. Bivariate correlations were evaluated to identify associations between BP measurements and the covariates. SBP, DBP, MAP were centred on 115, 90, 75 mmHg respectively, while age was centred on 55 years, BMI on 25 kg/m^2^, HDL-C on 1 mmol/L, and TC level on 2 mmol/L to facilitate interpretation.

Hierarchical linear regression models were used to quantify the association between BP levels (SBP, DBP, MAP) and brain volumes while controlling for age, education, diabetes mellitus, HDL-C, TC, antihypertensive medication, alcohol intake, physical activity, smoking status and intracranial volume (ICV) (Model 1). Two-way interactions between BP and age, and BP and antihypertensive medication were tested in a second model (Model 2). All analyses were stratified by sex as prior research has demonstrated sex variation [17], and pilot analyses indicated significant interactions between BP and sex. Analyses stratified by age (age ≤ 45 years) vs. (age 46–55 years) vs. (age 56–65 years) vs. (age > 65 years) were conducted as age is known to have differential effects on different blood pressure types. Furthermore, BMI effects were investigated further in post-hoc analyses as two-way interactions between BMI and BP in predicting brain volumes were identified in pilot analyses. Analyses controlling for PP as a marker of arterial stiffing were conducted as vascular stiffing may modulate the association between SBP and brain volumes. Sensitivity analyses strictly focused on normotensive participants were conducted to determine whether detected associations were also present below the clinical and pre-clinical thresholds.

Unstandardized beta coefficient and the corresponding *p* value were reported. A correction for multiple comparisons, statistical threshold was implemented. Nonlinear associations were explored by fitting a quadratic term for BP and age. Assumptions of linearity, including homoscedasticity and normality of residuals, were examined.

## 3. Results

Participants’ demographic and health characteristics are presented in Table 1. Compared to women, men had higher SBP (+5.4%) and DBP (+4.9%), and higher prevalence of hypertension (+14.8%), smoking (+7.1%), diabetes mellitus (+1.6%), and BMI (+4.2%). Pearson bivariate correlations between covariates are presented in Appendix A. Small to moderate associations were mostly observed between variables of interest, with the exception for BP measures that were highly inter-correlated (>0.74).

### 3.1. Age and Brain Volumes

After controlling for ICV, brain volumes were larger in men than women. However, greater age-related brain volume differences were observed in men. Every 1-year increase in age above 55 was associated with lower GMV (−0.31% vs. −0.20%), lower WMV (−0.32% vs. −0.16%), lower LHCV (−0.56% vs. −0.42%), and lower RHCV (−0.52% vs. −0.37%) in men compared to women. Scatter plots showing the associations between brain volumes and age are presented in the Appendix A.

### 3.2. BP and Brain Volumes

The associations between MAP, SBP, DBP and ROIs are presented in Table 2. Higher BP was associated with lower brain volumes in all ROIs (except for WMLs for which the association was reversed), with some variations across age groups and sex.

MAP was consistently negatively associated with GMV across age groups, but more so in women and in younger participants. The effect of DBP on GMV was greater (~59.8–61.9%) than that of SBP (Figure 1b,c). Every 10 mmHg increase in DBP above 75 mmHg was associated with a −040% (≤45 years) and −0.38% (46–55 years) lower GMV in women, and with a −0.40% (≤45 years) and −0.33% (46–55 years) lower GMV in men.

MAP was negatively associated with WMV, and more so in women and in younger participants. The effect of DBP was greater (~19–70%) than that of SBP on WMV (Figure 1b,c). Every 10 mmHg increase in DBP above 75 mmHg was associated with a −0.23% (≤45 years) and −0.24% (46–55 years) lower WMV in women, and with a 0.04% (≤45 years), and −0.23% (46–55 years) lower WMV in men. Note that the effects were non-significant in men (≤45 years).

MAP was negatively associated with HCV, except in men aged ≤45 years and women aged >65 years. The effect of DBP was greater (86.2–99%) than that of SBP (Figure 1b,c). Every 10 mmHg increase in DBP above 75 mmHg was associated with a −0.37% (≤45 years), and −0.17% (46–55 years) lower RHCV, and with −0.40% (≤45 years), and −0.10 (46–55 years) lower LHCV in women, and with a −0.37% (≤45 years), −0.20% (46–55 years) lower RHCV, and with 0.18% (≤45 years), and −0.10 (46–55 years) lower LHCV in men.

In addition, every 10 mmHg increase in DBP above 75 mmHg was associated with a 0.29% (>65 years) larger RHCV and with 0.34% (>65 years) larger LHCV in women. However, this association was negative in older men. Every 10 mmHg increase in DBP above 75 mmHg was associated with a −0.57% (>65 years) lower RHCV, and with −0.59% (>65 years) lower LHCV in men (Figure 1c).

Moreover, the effect of SBP was greater (~56.3–81.1%) than that of DBP on HCV in women than in men aged 56–65 years. Every 10 mmHg increase in SBP above 110 mmHg was associated with a −0.12% (56–65 years) lower RHCV, and −0.10% (56–65 years) lower LHCV in women; and a −0.008% (56–65 years) lower RHCV, and −0.07% (56–65 years) lower LHCV in men. However, the effect did not reach significance in men (Figure 1b).

Finally, MAP was consistently positively associated with WMLs across age groups, but more so in older participants. Every 10 mmHg increase in MAP above 90 mmHg was associated with a 1.23% (>65 years) larger WMLs in women and with 1.19% (>65 years) larger WMLs in men (Figure 1a).

#### BMI Effects

Detailed analyses of BMI and its interactions with BP are reported in Table 3. A significant two-way interaction was detected between BMI and BP with some notable differences between BP types, age groups and sex. Therefore, further analyses stratified by BMI category (normal: BMI ≤ 25; overweight/obese: BMI ≥ 25) were conducted.

The negative association between MAP and GMV was stronger in men with overweight/obese compared to normal BMI at ages ≤ 45 years. Every 10 mmHg increase in MAP above 90 mmHg was associated with a further 0.04% smaller GMV in those with overweight/obesity. Similar effects were not detected in women.

The positive association between MAP and WMLs was greater in women with overweight/obese compared to normal BMI between 56–65 years. Every 10 mmHg increase in MAP above 90 mmHg was associated with a further 0.04% larger WMLs in those with overweight/obesity. Similar effects were not detected in men. There were no interaction effects between BMI and BP in women.

### 3.3. Antihypertensive Medication Effects

Detailed analyses of antihypertensive medication and its interactions with BP are reported in Appendix A. Antihypertensive medication was found to significantly modulate the association between BP and brain volumes with some notable differences between BP types across age groups and sex.

In those ≤ 45 years, every 10 mmHg increase in SBP was associated with 0.9% less decline in GMV in men treated for hypertension but not in women. 

Every 10 mmHg increase in DBP was associated with 0.92% less decline in WMV in treated women. This was not observed in men. Moreover, every 10 mmHg increase in SBP was associated with 2.0% and 2.4% larger RHCV and LHCV in treated men but not in women.

In those >65 years, every 10 mmHg increase in SBP and DBP was associated with −1.2% and −1.6% lower WMLs in women treated for hypertension but not in men. No antihypertensive medication effects were detected between 45–55 and 56–65 years.

### 3.4. Vascular Stiffness Effects

Detailed analysis on the association between SBP and brain volumes after controlling for PP (as a marker of vascular stiffness) are presented in Appendix A. Although the association pattern was consistent with that reported above, some findings became more significant. Specifically, the association between SBP and HCV and WMLs were stronger in older age (>65 years), and across all age groups for GMV after PP adjustment (Figure 2).

### 3.5. Effects within the Normal BP Range

Sensitivity analyses limited to those with BP within the normal range were conducted to confirm that the major effects presented above were not uniquely attributable to hypertension or pre-hypertension. Higher MAP was significantly associated with lower GMV and larger WMLs within the normal MAP range in both sexes across age groups, although the effect was weaker in men. In addition, higher DBP was significantly associated with lower HCV (≤45 years) in women, and with RHCV (46–55 years) and LHCV (56–65 years) in men (Table 4).

## 4. Discussion

This study produced three important findings. First, it revealed that higher BP was consistently associated with smaller brain volumes with marked regional differences, and larger WMLs. Second, this effect was observed across the full BP range. Finally, stronger effects were detected in women than men, and in younger individuals.

The key finding is that higher BP was associated with poorer cerebral health in all brain regions and WMLs. These effects were particularly substantial for GMV and WMLs in people in their forties, although they were observed across all age groups. Indeed, compared to an individual with an optimal BP at age ≤ 45 years (e.g., SBP = 115, DBP = 75, MAP = 90 mmHg), a person with a higher BP (e.g., SBP = 140, DBP = 90, MAP = 110 mmHg) was predicted to have a ~0.7% lower GMV and ~0.8% larger WMLs. This effect is equivalent to approximately 1–2 years of aging in this age group and can therefore be considered clinically important [27]. Similar, although somewhat smaller, effects were seen for total WMV and other brain regions. Thus, some brain regions (e.g., GMV) appear to be more vulnerable to BP-related neurodegeneration as well as to increased WMLs.

Another notable finding is that a strong association was detected between BP and HCV. This is not surprising as the hippocampus is particularly vulnerable to environmental stressors [28], and a negative effect of high BP on this structure has already been demonstrated in the elderly [5]. However, it is particularly interesting that although this association was found in all age groups, it was non-linear and peaked in the forties. It then decreased progressively in both sexes and surprisingly reversed direction above 65 years, but in women only. The reason for this positive association is unclear, but it is possible that it reflects a transient effect of inflammatory processes leading to a temporary volumetric increase [29]. It is also particularly noteworthy that the non-linear associations were mainly driven by DBP suggesting that both hypo-perfusion and hypertension related to DBP is associated with brain damage. Consistent with this view are recent findings that demonstrated a J-shape association between low DBP and an increased risk of cardiovascular events [30]. Therefore, targeting an optimum DBP level is important to ensure adequate blood perfusion to sensitive organs such as the brain, and to protect against cognitive decline.

These findings also imply that the effects of MAP compared to SBP or DBP differ (Figure 1). For example, the effects of a rising BP have a different magnitude depending on the type of BP measure (SBP, MAP, or DBP) considered with important implications for clinical BP management.

### Implications for Health Policy and Clinical Practice

Of particular relevance to health policy and clinical guidelines, the associations between higher BP and impaired cerebral health were detected across the full BP range and were still detectable in those with normal BP and not treated with anti-hypertensive medication. This is consistent with the previous findings from a recent systematic review [5], whose sensitivity analyses indicated that the deleterious effect of higher MAP on WMLs and GMV presents not only in hypertension and prehypertension categories but also in the normal MAP range and in all age groups, although weaker associations were observed in men. The reason for these sex differences may be due to physiological differences or differential exposure to risk factors in men and women. For example, the reduction in estrogen levels after menopause, which would be expected to reduce the neuroprotective effect of estrogens [31], and higher prevalence of midlife obesity (15%) in women [32], are factors that increase women’s vulnerability to high BP-related neurodegeneration across the life-course.

One important question raised by these findings is whether the variability of observed effects in different age groups is attributable to the progressive arterial stiffening typically observed in older individuals and due to vascular calcification associated with the aging process [33]. PP is a practical index of arterial stiffening in the present context, and we used it in sensitivity analyses to assess the possible impact of arterial stiffening. Individuals with greater stiffening (i.e., higher PP) were more likely to have smaller GMV, HCV and larger WMLs, particularly in those individuals with hypertension. Thus, elevated BP associated with vascular stiffening may lead to inadequate cerebral perfusion, leading to apoptosis, neurodegeneration and brain shrinkage [34].

The adverse effect of high BP was also found to be worse at higher BMI. On average, those individuals with overweight/obesity experienced an additional ~0.4% lower GMV for every 10 years above 45 in men. The fact that higher BMI is related to worse cerebral health is not surprising [35,36,37], but it is interesting that this effect is more apparent in men in their forties than in women. However, a greater effect was found in women aged between 56–65 years for WML. It seems that the risk of obesity on higher BP-related brain shrinkage is lower in women than men at younger ages (<~50 years), but is reversed post-menopause. This may suggest that women are less protected against the effects of lifestyle factors on high BP-related neurodegeneration when they reach menopause.

Another important finding is that anti-hypertensive medication use appears to be protective for GMV, WMV and HCV in younger participants. These findings may partly explain the association between medication use and decreased dementia risk demonstrated in a recent meta-analysis [38], which showed that individuals who are treated in their forties are less vulnerable to BP-related brain damage. These findings may have important clinical and population health implications as they suggest that controlling BP at younger ages may make a greater contribution to reducing the risk of dementia in later life.

Furthermore, it is noteworthy that only 7.2% of individuals with hypertension (41.2%) in our sample were treated with anti-hypertensive medication. This is low, but unfortunately it is also consistent with many findings in the literature. Indeed, Muller et al., (2014) reported that 35% of their sample had hypertension, while only 6% were taking antihypertensive medication [39]. Lane et al., (2019) determined that the prevalence of hypertension in their cohort was 16% while only 2% used antihypertensive treatment [11]. Moreover, Caunca et al., (2020) reported that 82% of the population they studied had an SBP: 130–139 mmHg or DBP: 80–89 mmHg, but only 40% of those was treated [40]. In addition, a recent WHO report (2021) indicated that more than half of individuals with hypertension worldwide (720 million people), are not receiving any BP medication [41]. This further highlight the pressing need for educational and public health interventions aimed at increasing detection in the population, particularly in early to mid-adulthood, and to better communicate the benefits, both cardio-vascular and neurocognitive, of maintaining a healthy, if not optimal, blood pressure.

This study has a number of limitations but also significant strengths. It investigated participants in middle-age to early old age, and therefore findings may not be generalisable to early adulthood (<40 years) or older adulthood (>80 years). Secondly, while the overall sample was very large, analyses were stratified by age and sex, leading to somewhat uneven cell sizes. This may have reduced our capacity to detect a certain effect in some age groups (e.g., >65 years). However, even in this age group the sample size was relatively large and, consequently, unlikely to have substantially influenced the broad pattern of results. Other limitations include BP being assessed in a single session (although averaged over two measurements) by peripheral measure, which is known to be imprecise. Thus, future studies should confirm these findings using central measures of BP. Although we have carefully controlled for the possible effects of some major cardiovascular, lifestyle, and socio-demographic risk factors in our analyses it is likely that this did not account fully for their influence or for the effects of other uncontrolled risk factors on brain structure (e.g., atherosclerosis, valvular heart disease). Consequently, future research should aim to include and contrast a greater variety of cardiovascular measures to achieve a more precise assessment of the effects of cardiac and large vessels conditions on brain health. Finally, the cross-sectional nature of this study does not allow us to draw conclusions on causation or directionality. Thus, longitudinal studies are needed to investigate these questions. In contrast, the study includes a large sample size that provides sufficient power to assess associations and interactions in detail, and to control for a wide range of covariates. It only included cognitively healthy participants and thus, it is unlikely to mainly reflect the consequences of clinical neurological conditions. Finally, participants were scanned using the same method (scanner, magnetic field, and parameters), and images were segmented with a semi-automated technique, thus minimising possible sources of bias.

## 5. Conclusions

In conclusion, this community-based study of participants aged 40 to 70 years shows that elevated BP across its full spectrum is linked to worse cerebral health in all ages studied, demonstrating that the onset of BP-related risk of premature neurodegeneration occurs as early as 40 years. Our findings provide support for clinical guidelines to recommend preventative measures for younger populations with a stronger emphasis on achieving optimal BP levels.

## Figures and Tables

**Figure 1 jcm-11-03127-f001:**
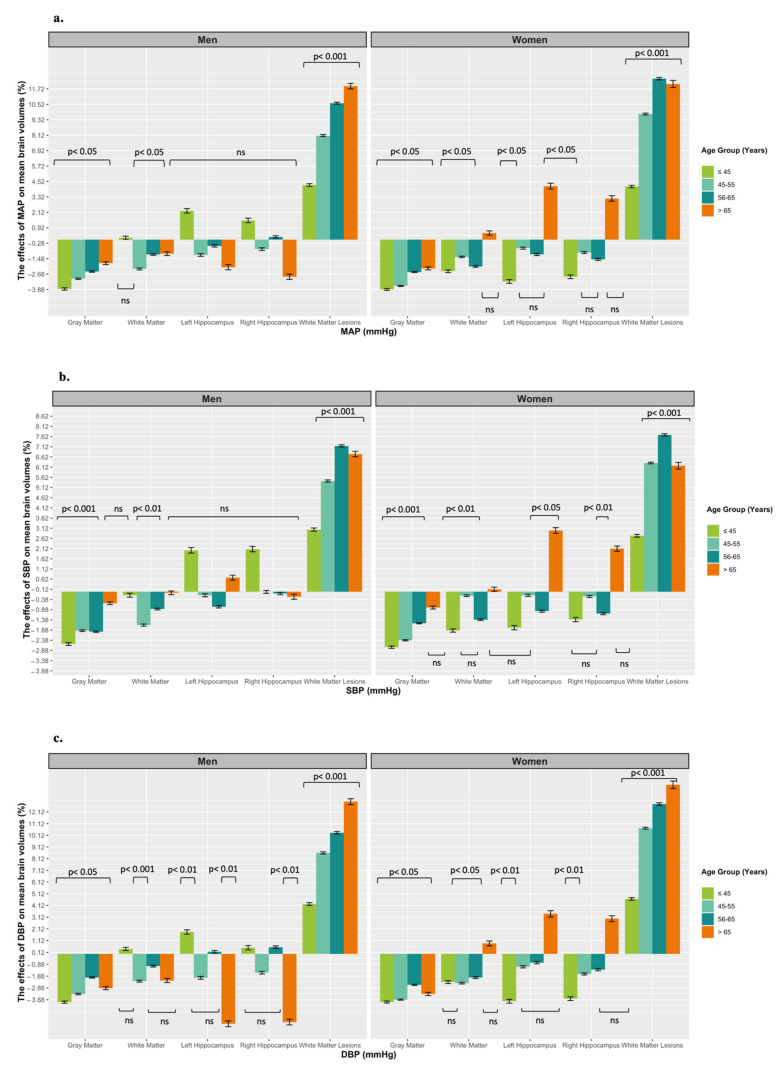
The proportion effects of BP including (**a**) mean arterial pressure (MAP), (**b**) systolic blood pressure (SBP), (**c**) diastolic blood pressure (DBP), on the brain volumes including gray matter, white matter, left and right hippocampus and white matter lesions, across age groups in men and women. Association presented as unstandardized beta coefficient and corresponding standard error (SD). Note that for ease of interpretation, we exponentiated the coefficients to produce the proportionate effect of a 10 mmHg higher blood pressure on mean brain volume for each brain region and white matter lesion (WMLs) in men and women. For instance, an exponentiated coefficient of 1.1 reflects a 10% increase in mean brain volume.

**Figure 2 jcm-11-03127-f002:**
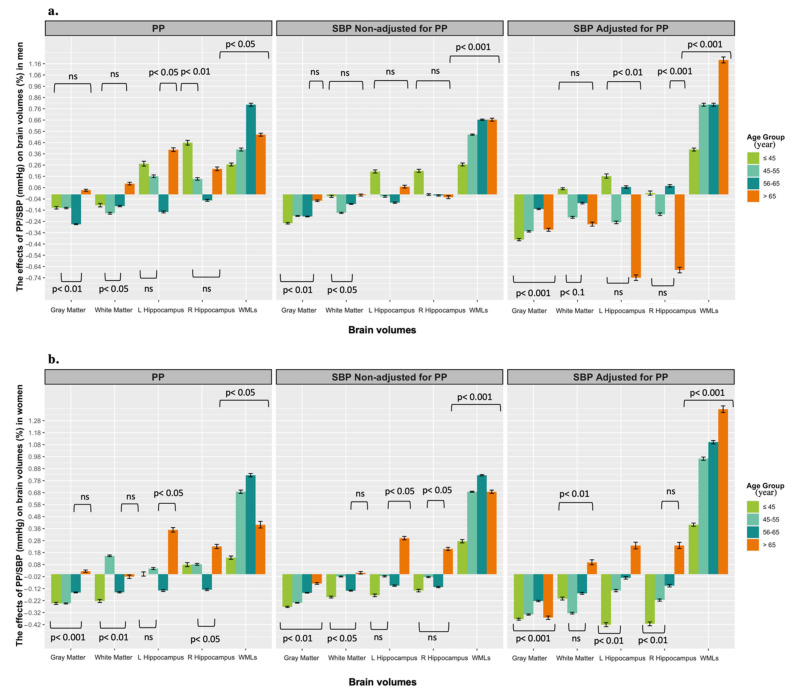
Proportion effects of pulse pressure (PP) and systolic blood pressure (SBP) on the brain volumes including gray matter, white matter, left hippocampus (L hippocampus), right hippocampus (R hippocampus), and white matter lesions (WMLs) across age groups in men (**a**) and women (**b**). Graphs compare the effects of PP, SBP non-adjusted by PP and SBP adjusted by PP for each brain region and WMLs in men and women. Association presented as unstandardized beta coefficient and corresponding standard error (SD). Note that for ease of interpretation, we exponentiated the coefficients to produce the proportionate effect of a 10 mmHg higher blood pressure on mean brain volume for each brain region and WMLs in men and women. For instance, an exponentiated coefficient of 1.1 reflects a 10% increase in mean brain volume.

**Table 1 jcm-11-03127-t001:** Participants’ demographic characteristics.

Measures	Whole Sample	Men	Women	T/chi-sq Test (*p*-Value)
Age, year (SD)	54.88 (7.47)	55.58 (7.57)	54.26 (7.33)	16.89 (0.000)
SBP, mmHg (SD)	134.90 (17.71)	138.80 (16.46)	131.39 (18.06)	40.85 (0.000)
DBP, mmHg (SD)	81.37 (9.90)	83.56 (9.64)	79.41 (9.72)	40.79 (0.000)
MAP, mmHg (SD)	99.21 (11.66)	101.97 (11.07)	96.73 (11.63)	43.91 (0.000)
PP, mmHg (SD)	53.52 (12.34)	55.24 (11.55)	51.98 (12.80)	25.45 (0.000)
GM, mm^3^ (SD)	665,373.52 (59,587.27)	699,375.78 (53,696.16)	634,797.84 (46651.03)	121.62 (0.000)
WM volume, mm^3^ (SD)	478,100.24 (57,260.97)	508,577.33 (53,010.15)	450,694.48 (45979.19)	110.50 (0.000)
Left HC volume, mm^3^ (SD)	3673.24 (395.67)	3689.78 (389.09)	3649.69 (403.70)	9.45 (0.000)
Right HC volume, mm^3^ (SD)	3792.48 (402.87)	3807.18 (394.35)	3771.55 (413.81)	8.23 (0.000)
WMLs volume, mm^3^ (SD)	7.40 (0.68)	7.54 (0.67)	7.27 (0.66)	37.42 (0.000)
ICV, mm^3^ (SD)	1,548,411.25 (152,104.52)	1,542,329.21 (149,601.04)	1,557,069.86 (155,193.42)	−9.04 (0.000)
BMI, kg/m^2^ (SD)	26.56 (4.22)	27.12 (3.76)	26.07 (4.54)	24.16 (0.000)
Cholesterol, mmol/L, (SD)	5.73 (1.08)	5.59 (1.08)	5.85 (1.07)	−22.50 (0.000)
HDL mmol/L, (SD)	1.48 (0.38)	1.31 (0.30)	1.63 (0.37)	−91.50 (0.000)
Hypertension, *n* (%)	14,961 (41.26%)	8421 (49.05%)	6540 (34.26%)	815.84 (0.000)
Antihypertensive medication, *n* (%)	2645 (7.29%)	1334 (7.77%)	1311 (6.87%)	10.78 (0.001)
Diabetes mellitus, *n* (%)	950 (2.62%)	473 (2.22%)	477 (3.19%)	31.87 (0.000)
Ever smoked, *n* (%)	14,149 (39.02%)	7979 (37.46%)	6170 (41.24%)	52.58 (0.000)
Higher Education, *n* (%)	17,525 (48.33%)	10,692 (50.20%)	6833 (45.67%)	71.95 (0.000)

Significance: *p* < 0.05. Abbreviations: SD, standard deviation; *n*, number of participants; SBP, systolic blood pressure; DBP, diastolic blood pressure; MAP, main arterial pressure; PP, pulse pressure; GM, gray matter; WM, white matter; HC, hippocampus; WMLs, white matter lesions; ICV, intracranial volume; BMI, body mass index; HDL, high-density lipoprotein.

**Table 2 jcm-11-03127-t002:** Association between baseline BP including MAP, SBP, and DBP and brain volumes across age groups in men and women at UK biobank study.

	Gray Matter Volume (mm^3^)		White Matter Volume (mm^3^)		Left Hippocampus Volume (mm^3^)		Right Hippocampus Volume (mm^3^)		White Matter Lesions Volume (mm^3^)	
	Beta (SE)		Beta (SE)		Beta (SE)		Beta (SE)		Beta (SE)	
	**Participants Aged ≤ 45 Years**									
	**Men, *n* = 2220**	**Women,** ***n* = 2867**	**Men, *n* = 2220**	**Women, *n* = 2867**	**Men, *n* = 2220**	**Women, *n* = 2867**	**Men,** ***n* = 2220**	**Women, *n* = 2867**	**Men, *n* = 2220**	**Women, *n* = 2867**
MAP	−268.346 *** (62.485)	−246.073 *** (47.730)	7.341 (63.676)	−110.942 ** (47.813)	0.850 (0.669)	−1.158 ** (0.519)	0.586 (0.677)	−1.061 ** (0.516)	0.003 *** (0.001)	0.003 *** (0.001)
SBP	−179.415 *** (47.880)	−172.891 *** (35.186)	−8.685 (48.744)	−85.832 ** (35.229)	0.773 (0.512)	−0.628 (0.383)	0.823 (0.518)	−0.502 (0.381)	0.002 *** (0.001)	0.002 *** (0.001)
DBP	−285.528 *** (66.856)	−258.127 *** (52.955)	21.088 (68.126)	−107.449 ** (53.033)	0.704 (0.716)	−1.426 ** (0.576)	0.203 (0.725)	−1.389 ** (0.572)	0.003 *** (0.001)	0.003 *** (0.001)
	**Participants Aged between 46–55 years**									
	**Men, *n* = 5699**	**Women,** ***n* = 7456**	**Men, *n* = 5699**	**Women, *n* = 7456**	**Men, *n* = 5699**	**Women, *n* = 7456**	**Men,** ***n* = 5699**	**Women, *n* = 7456**	**Men, *n* = 5699**	**Women, *n* = 7456**
MAP	−212.574 *** (38.583)	−228.546 *** (26.449)	−115.987 *** (37.605)	−60.878 ** (27.318)	−0.453 (0.394)	−0.236 (0.284)	−0.291 (0.399)	−0.368 (0.281)	0.006 *** (0.001)	0.007 *** (0.0005)
SBP	−133.847 *** (27.786)	−151.197 *** (18.091)	−83.331 *** (27.064)	−8.702 (18.685)	−0.067 (0.284)	−0.062 (0.194)	−0.003 (0.287)	−0.086 (0.192)	0.004 *** (0.0004)	0.005 *** (0.0003)
DBP	−237.579 *** (43.185)	−245.744 *** (30.831)	−117.177 *** (42.096)	−111.271 *** (31.804)	−0.770 * (0.441)	−0.390 (0.331)	−0.543 (0.447)	−0.625 * (0.327)	0.007 *** (0.001)	0.008 *** (0.001)
	**Participants Aged between 56–65 years**									
	**Men, *n* = 7708**	**Women,** ***n* = 7692**	**Men, *n* = 7708**	**Women, *n* = 7692**	**Men, *n* = 7708**	**Women, *n* = 7692**	**Men,** ***n* = 7708**	**Women, *n* = 7692**	**Men, *n* = 7708**	**Women, *n* = 7692**
MAP	−174.006 *** (33.094)	−160.135 *** (28.472)	−59.050 * (33.175)	−94.255 *** (28.835)	−0.185 (0.349)	−0.410 (0.305)	0.082 (0.356)	−0.561 * (0.306)	0.008 *** (0.001)	0.009 *** (0.001)
SBP	−137.254 *** (21.699)	−98.180 *** (17.826)	−43.220 ** (21.768)	−61.805 *** (18.050)	−0.280 (0.229)	−0.339 * (0.191)	−0.034 (0.233)	−0.394 ** (0.191)	0.005 *** (0.0004)	0.006 *** (0.0004)
DBP	−139.247 *** (38.773)	−166.069 *** (34.241)	−52.596 (38.834)	−90.344 *** (34.667)	0.065 (0.409)	−0.263 (0.366)	0.222 (0.416)	−0.489 (0.367)	0.008 *** (0.001)	0.009 *** (0.001)
		**Participants Aged > 65 years**								
	**Men, *n* = 1541**	**Women,** ***n* = 1077**	**Men, *n* = 1541**	**Women, *n* = 1077**	**Men, *n* = 1541**	**Women, *n* = 1077**	**Men,** ***n* = 1541**	**Women, *n* = 1077**	**Men, *n* = 1541**	**Women, *n* = 1077**
MAP	−128.205 * (75.395)	−142.297 * (73.217)	−55.493 (73.421)	23.048 (76.931)	−0.815 (0.763)	1.481 * (0.799)	−1.134 (0.794)	1.176 (0.774)	0.009 *** (0.002)	0.009 *** (0.002)
SBP	−39.328 (46.797)	−49.280 (45.713)	−2.706 (45.547)	5.374 (47.974)	0.263 (0.473)	1.070 ** (0.498)	−0.096 (0.493)	0.776 (0.482)	0.005 *** (0.001)	0.005 *** (0.001)
DBP	−202.744 ** (90.283)	−215.163 ** (87.574)	−114.477 (87.948)	39.658 (92.110)	−2.245 ** (0.913)	1.212 (0.957)	−2.264 ** (0.951)	1.098 (0.927)	0.010 *** (0.002)	0.010 *** (0.002)

Significance. * *p* < 0.05; ** *p* < 0.01; *** *p* < 0.001. Abbreviations: *n*, number of participants; SE, standard error; MAP, mean arterial pressure; SBP, systolic blood pressure; DBP, diastolic blood pressure. Hierarchical linear regression models were used to quantify the correlation (Beta, and SE) between BP levels (MAP, SBP, and DBP) and brain volumes while adjusting for total intracranial volume, and main covariates: HDL (high-density lipoprotein), cholesterol, diabetes, smoking status, higher education, physical activity, alcohol intake, and antihypertensive medication in model 1. Note that for ease of interpretation, we exponentiated the coefficients to produce the proportionate effect of a 10 mm Hg higher blood pressure on mean brain volume for each brain region and white matter lesions (WMLs) in men and women. For instance, an exponentiated coefficient of 1.1 reflects a 10% increase in mean brain volume.

**Table 3 jcm-11-03127-t003:** Interaction between baseline BP including MAP, BMI, and brain volumes across age groups in men and women at UK biobank study.

	Gray Matter Volume (mm^3^)		White Matter Volume (mm^3^)		Left Hippocampus Volume (mm^3^)		Right Hippocampus Volume (mm^3^)		White Matter Lesions Volume (mm^3^)	
	Beta (SE)		Beta (SE)		Beta (SE)		Beta (SE)		Beta (SE)	
	**Participants Aged ≤ 45 Years**									
	**Men*, n* = 2220**	**Women,** ***n* = 2867**	**Men, *n* = 222**	**Women, *n* = 2867**	**Men, *n* = 2220**	**Women, *n* = 2867**	**Men,** ***n* = 2220**	**Women, *n* = 2867**	**Men, *n* = 2220**	**Women, *n* = 2867**
MAP	−198.625 *** (66.470)	−208.734 *** (50.092)	79.340 (67.656)	−59.634 (50.123)	0.960 (0.713)	−1.165 ** (0.545)	0.644 (0.722)	−1.145 ** (0.542)	0.002 ** (0.001)	0.003 *** (0.001)
BMI	16.382 (239.070)	−275.082 ** (134.932)	−633.982 *** (243.336)	−365.703 *** (135.014)	−0.573 (2.564)	0.367 (1.469)	−0.718 (2.596)	0.873 (1.460)	0.006 ** (0.003)	0.003 (0.002)
MAPxBMI	−29.473 ** (13.137)	7.608 (8.625)	−6.065 (13.371)	−2.559 (8.630)	−0.056 (0.141)	−0.039 (0.094)	0.011 (0.143)	−0.051 (0.093)	0.0003 (0.0002)	0.0001 (0.0001)
	**Participants Aged between 46–55 years**									
	**Men, *n* = 5699**	**Women,** ***n* = 7456**	**Men, *n* = 5699**	**Women, *n* = 7456**	**Men, *n* = 5699**	**Women, *n* = 7456**	**Men,** ***n* = 5699**	**Women, *n* = 7456**	**Men, *n* = 5699**	**Women, *n* = 7456**
MAP	−192.823 *** (40.704)	−200.841 *** (27.180)	−85.433 ** (39.624)	0.769 (27.990)	−0.644 (0.417)	−0.320 (0.293)	−0.464 (0.422)	−0.467 (0.290)	0.006 *** (0.001)	0.008 *** (0.001)
BMI	−596.787 *** (168.370)	−341.246 *** (89.410)	−769.786 *** (163.904)	−628.097 *** (92.074)	1.111 (1.724)	−0.613 (0.963)	2.921 * (1.744)	−0.319 (0.952)	0.005 * (0.003)	−0.001 (0.002)
MAP×BMI	4.411 (9.513)	−4.452 (5.326)	5.635 (9.260)	2.521 (5.485)	−0.029 (0.097)	−0.020 (0.057)	−0.113 (0.099)	−0.018 (0.057)	0.0001 (0.0001)	0.0001 (0.0001)
	**Participants Aged between 56–65 years**									
	**Men, *n* = 7708**	**Women,** ***n* = 7692**	**Men, *n* = 7708**	**Women, *n* = 7692**	**Men, *n* = 7708**	**Women, *n* = 7692**	**Men,** ***n* = 7708**	**Women, *n* = 7692**	**Men, *n* = 7708**	**Women, *n* = 7692**
MAP	−181.194 *** (34.689)	−138.683 *** (28.930)	−9.496 (34.638)	−37.344 (29.165)	−0.333 (0.366)	−0.587 * (0.310)	−0.079 (0.373)	−0.582 * (0.311)	0.008 *** (0.001)	0.010 *** (0.001)
BMI	−404.863 ** (160.237)	−387.165 *** (107.476)	−714.330 *** (160.000)	−868.234 *** (108.351)	0.522 (1.691)	0.770 (1.152)	1.385 (1.721)	−0.853 (1.155)	0.009 *** (0.003)	0.001 (0.002)
MAP×BMI	8.606 (8.686)	−7.511 (6.489)	−9.864 (8.674)	4.697 (6.541)	0.010 (0.092)	0.008 (0.070)	−0.020 (0.093)	0.033 (0.070)	0.0002 (0.0002)	0.0003 * (0.0002)
		**Participants Aged > 65 years**								
	**Men, *n* = 1541**	**Women,** ***n* = 1077**	**Men, *n* = 1541**	**Women, *n* = 1077**	**Men, *n* = 1541**	**Women, *n* = 1077**	**Men,** ***n* = 1541**	**Women, *n* = 1077**	**Men, *n* = 1541**	**Women, *n* = 1077**
MAP	−111.270 (77.719)	−153.558 ** (73.907)	31.635 (75.217)	52.849 (77.419)	−0.833 (0.787)	1.181 (0.806)	−0.902 (0.818)	0.968 (0.780)	0.010 *** (0.002)	0.010 *** (0.002)
BMI	27.947 (395.892)	−199.765 (325.092)	−765.763 ** (383.148)	−606.394 * (340.537)	2.249 (4.007)	4.827 (3.544)	5.229 (4.167)	1.386 (3.430)	0.008 (0.008)	−0.011 (0.009)
MAP×BMI	−9.049 (22.063)	18.236 (18.549)	−27.683 (21.353)	0.783 (19.430)	−0.034 (0.223)	−0.173 (0.202)	−0.331 (0.232)	0.161 (0.196)	−0.0002 (0.0005)	0.001 (0.0005)

Significance. * *p* < 0.05; ** *p* < 0.01; *** *p* < 0.001. Abbreviations: *n*, number of participants; SE, standard error; MAP, mean arterial pressure; BMI, body mass index. Hierarchical linear regression models were used to quantify the correlation (Beta, and SE) between MAP and brain volumes while adjusting for basic covariates: total intracranial volume, and main covariates: HDL (high-density lipoprotein), cholesterol, diabetes, smoking status, higher education, physical activity, alcohol intake, antihypertensive medication and BMI and tested for the statistical interactions between (BP × BMI) in predicting brain volume measures in model 3. Note that for ease of interpretation, we exponentiated the coefficients to produce the proportionate effect of a 10 mm Hg higher blood pressure on mean brain volume for each brain region and WMLs in men and women. For instance, an exponentiated coefficient of 1.1 reflects a 10% increase in mean brain volume.

**Table 4 jcm-11-03127-t004:** Association between baseline normal BP including MAP, SBP, and DBP level and brain volumes across age groups in men and women at UK biobank study.

	Grey Matter Volume (mm^3^)		White Matter Volume (mm^3^)		Left Hippocampus Volume (mm^3^)		Right Hippocampus Volume (mm^3^)		White Matter Lesions Volume (mm^3^)	
	Beta (SE)		Beta (SE)		Beta (SE)		Beta (SE)		Beta (SE)	
	**Participants Aged ≤ 45 Years**									
	**Men,** ***n* = 964**	**Women, *n* = 1987**	**Men,** ** *n* ** **= 964**	**Women, *n* = 1987**	**Men,** ***n* = 964**	**Women, *n* = 1987**	**Men,** ** *n* ** **= 964**	**Women, *n* = 1987**	**Men,** ***n* = 964**	**Women, *n* = 1987**
Normal MAP range	−140.655 (183.520)	−243.201 ** (96.291)	263.209 (190.018)	−95.851 (97.375)	2.359 (1.942)	−2.016 * (1.038)	1.640 (1.945)	−1.682 (1.028)	0.002 (0.002)	0.0003 (0.001)
Normal SBP range	−42.742 (106.502)	−164.736 *** (62.863)	23.849 (110.357)	−92.306 (63.560)	1.444 (1.126)	−0.456 (0.679)	1.234 (1.128)	−0.376 (0.672)	0.002 (0.001)	0.00005 (0.001)
Normal DBP range	−186.480 (206.326)	−194.894 * (102.694)	454.374 ** (213.363)	−40.338 (103.798)	1.762 (2.184)	−2.828 ** (1.106)	0.793 (2.187)	−2.366 ** (1.095)	−0.001 (0.002)	0.0004 (0.001)
	**Participants Aged between 46–55 years**									
	**Men, *n* = 1980**	**Women, *n* = 4043**	**Men, *n* = 1980**	**Women, *n* = 4043**	**Men, *n* = 1980**	**Women, *n* = 4043**	**Men, *n* = 1980**	**Women, *n* = 4043**	**Men, *n* = 1980**	**Women, *n* = 404**
Normal MAP range	−31.049 (127.963)	−128.719 ** (63.733)	−241.406 * (124.627)	−35.402 (64.560)	1.144 (1.352)	−0.151 (0.685)	2.084 (1.381)	0.357 (0.681)	−0.001 (0.002)	0.005 *** (0.001)
Normal SBP range	5.250 (70.961)	−87.950 ** (36.926)	−161.276 ** (69.080)	33.839 (37.410)	1.030 (0.750)	−0.058 (0.397)	1.435 * (0.765)	0.378 (0.394)	−0.0001 (0.001)	0.003 *** (0.001)
Normal DBP range	−71.526 (145.776)	−80.381 (72.517)	−129.634 (142.087)	−133.879 * (73.404)	0.054 (1.541)	−0.181 (0.779)	1.029 (1.574)	−0.035 (0.774)	−0.001 (0.002)	0.005 **** (0.001)
	**Participants Aged between 56–65 years**									
	**Men, *n* = 2499**	**Women, *n* = 3695**	**Men, *n* = 2499**	**Women, *n* = 3695**	**Men, *n* = 2499**	**Women, *n* = 3695**	**Men, *n* = 2499**	**Women, *n* = 3695**	**Men, *n* = 2499**	**Women, *n* = 3695**
Normal MAP range	−196.624 * (104.472)	−190.525 *** (69.524)	69.218 (107.730)	−73.460 (70.536)	−0.417 (1.118)	−0.387 (0.751)	0.339 (1.132)	−0.147 (0.749)	0.008 *** (0.002)	0.008 *** (0.001)
Normal SBP range	−133.036 *** (51.358)	−100.866 *** (34.649)	20.373 (52.996)	−60.706 * (35.149)	−0.243 (0.550)	−0.226 (0.375)	0.194 (0.557)	−0.126 (0.373)	0.005 *** (0.001)	0.005 *** (0.001)
Normal DBP range	−32.864 (133.890)	−133.617 (88.385)	101.246 (137.965)	19.338 (89.620)	−0.201 (1.432)	−0.203 (0.955)	0.175 (1.449)	0.053 (0.952)	0.003 (0.002)	0.005 *** (0.002)
	**Participants Aged > 65 years**									
	**Men,** ***n* = 560**	**Women, *n* = 512**	**Men,** ***n* = 560**	**Women, *n* = 512**	**Men,** ***n* = 560**	**Women, *n* = 512**	**Men,** ***n* = 560**	**Women, *n* = 512**	**Men,** ***n* = 560**	**Women, *n* = 512**
Normal MAP rang	−190.065 (218.158)	−360.197 * (183.829)	302.068 (217.392)	−82.686 (185.861)	3.453 (2.292)	0.497 (1.892)	2.914 (2.291)	−1.511 (1.782)	0.011 *** (0.004)	0.011 ** (0.004)
Normal SBP range	−107.070 (103.988)	−94.429 (90.647)	124.976 (103.697)	−59.590 (91.376)	1.821 * (1.092)	0.334 (0.930)	1.660 (1.092)	−0.797 (0.876)	0.007 *** (0.002)	0.004 * (0.002)
Normal DBP range	−90.755 (295.256)	−625.745 ** (246.919)	326.066 (294.230)	−1.479 (250.343)	2.149 (3.105)	0.099 (2.548)	1.318 (3.103)	−1.121 (2.401)	0.002 (0.006)	0.013 ** (0.006)

Significance. * *p* < 0.05; ** *p* < 0.01; *** *p* < 0.001. Abbreviations: *n*, number of participants; SE, standard error; MAP, mean arterial pressure; SBP, systolic blood pressure; DBP, diastolic blood pressure. Sensitivity analyses limited to those with blood pressure (BP) within the normal (SBP < 120 or DBP < 80 mmHg). Hierarchical linear regression models were used to quantify the correlation (Beta, and SE) between normal BP (MAP, SBP, and DBP) and brain volumes while adjusting for basic covariates: total intracranial volume, and main covariates: HDL (high-density lipoprotein), cholesterol, diabetes, smoking status, higher education, physical activity, alcohol intake, and antihypertensive medication. Note that for ease of interpretation, we exponentiated the coefficients to produce the proportionate effect of a 10 mm Hg higher blood pressure on mean brain volume for each brain region and white matter lesions (WMLs) in men and women. For instance, an exponentiated coefficient of 1.1 reflects a 10% increase in mean brain volume.

## Data Availability

UK Biobank is an open access resource accessible to confirmed researchers upon request (ukbiobank.ac.uk/).

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
