# Peer review of "Effects of Higher Normal Blood Pressure on Brain Are Detectable before Middle-Age and Differ by Sex"

_jcm, 2022, doi:10.3390/jcm11113127_

Round 1
Reviewer 1 Report
I have a general concern about white matter lesions being referred to as a brain region. I know it is easier to write the paper this way, but WMLs are not a brain region; they are an effect of injury that can be diverse and inconsistent.
Were the patients assessed for cognitive dysfunction as they aged?
The discrepancy between the percentage of the population with hypertension and the rate that is being treated was surprising. Can the authors please comment?
What is known about blood pressure control in the population treated for hypertension?
Author Response
We would like to thank this reviewer for their comments, and we appreciate the time and effort that they have taken to review our manuscript. We have addressed all the points they raised below and throughout the manuscript.
Reviewer 1 (Point 1)
I have a general concern about white matter lesions being referred to as a brain region. I know it is easier to write the paper this way, but WMLs are not a brain region; they are an effect of injury that can be diverse and inconsistent.
Response (Point 1)
Thank you for alerting us to this inaccurate use of the words “region” when referring to white matter lesions. We agree that white matter lesions are not a brain region; rather, they are a result of an injury that can be varied and inconsistent. We have edited the manuscript throughout to consistently refer to them as white matter lesions or WMLs.
The changes were indicated by the track changes in the manuscript
Reviewer 1 (Point 2)
Were the patients assessed for cognitive dysfunction as they aged?
Response (Point 2)
Yes, the cognitive function of UK Biobank participants was evaluated using neuropsychological tests, clinical evaluations, and information from their medical records. This information was considered to select participants for this study. All participants investigated were free of dementia and neurological diseases. This point has been described in method section (page 2, line 78-82).
Reviewer 1 (Point 3)
The discrepancy between the percentage of the population with hypertension and the rate that is being treated was surprising. Can the authors please comment? What is known about blood pressure control in the population treated for hypertension?
Response (Point 3)
Indeed, the disparity between the proportion of the population with hypertension (41.2%) and the number of participants receiving treatment (7.2%) may appear large. However, it is not inconsistent with findings in the literature which reports high levels of undetected or untreated hypertension in community-based samples [1–5]. For example, Muller et al., (2014) reported that 35% of their sample had hypertension, while only 6% were taking antihypertensive medication [1]. Lane et al., (2019) determined that the prevalence of hypertension in their cohort was 16% while only 2% used antihypertensive treatment [2]. Moreover, Caunca et al. (2020) reported that 82% of the population they studied had a SBP: 130–139 mmHg or DBP: 80–89 mmHg, but only 40% of those was treated [5]. In addition, a recent WHO report (2021) indicated that more than half of individuals with hypertension worldwide (720 million people), are not receiving any BP medication [6].
We have now commented on this issue in our manuscript as follows (page 17, line 365-378):
“It is noteworthy that only 7.2% of individuals with hypertension (41.2%) in our sample were treated with anti-hypertensive medication. This is low, but unfortunately it is also consistent with many findings in the literature. Indeed, Muller et al., (2014) reported that 35% of their sample had hypertension, while only 6% were taking antihypertensive medication [1]. Lane et al., (2019) determined that the prevalence of hypertension in their cohort was 16% while only 2% used antihypertensive treatment [2]. Moreover, Caunca et al. (2020) reported that 82% of the population they studied had a SBP: 130–139 mmHg or DBP: 80–89 mmHg, but only 40% of those was treated [5]. In addition, a recent WHO report (2021) indicated that more than half of individuals with hypertension worldwide (720 million people), are not receiving any BP medication [6]. This further highlight the pressing need for educational and public health interventions aimed at increasing detection in the population, particularly in early to mid-adulthood, and to better communicate the benefits, both cardio-vascular and neurocognitive, of maintaining a healthy if not optimal blood pressure.”

Reviewer 2 Report
Alateeq et al quantify the association between blood pressure (BP) across its full range and brain volumes and white matter lesions (WMLs) while investigating the effects of age, sex, body mass index (BMI), and antihypertensive medication. They concluded that higher BP is associated with worse cerebral health across the full BP range from middle adulthood and across the life course.
This is a very written manuscript and relevant finding for the general population and cerebral health.
Author Response
Reviewer 2 (Point 1)
Alateeq et al quantify the association between blood pressure (BP) across its full range and brain volumes and white matter lesions (WMLs) while investigating the effects of age, sex, body mass index (BMI), and antihypertensive medication. They concluded that higher BP is associated with worse cerebral health across the full BP range from middle adulthood and across the life course. This is a very written manuscript and relevant finding for the general population and cerebral health.
Response (Point 1)
We would like to thank the reviewer for their comments, and we appreciate the time and effort that they have taken to review our manuscript.

Reviewer 3 Report
An interesting work with high scientific and clinical value. It deals with the effects of blood pressure on the condition of the brain in terms of volumetric parameters such as total white matter volume, total gray matter volume, focal changes volume and the volume of the right and left hippocampuses.
It is based on a wide material and is analysed with well-selected statistical tools. The results are presented logically and clearly. The discussion is logical and communicative.
My only concern is the lack of information on possible cardiovascular risk factors. Neurological diseases are not the only ones responsible for the structural condition of the brain.
Pulse pressure alone does not explain the complexity of the effects of the heart and the large vessels state on the central nervous system; important are for example the condition of carotid arteries or the heart valves. Please refer to this suggestion in the methodology or write it in the limits of the study.
Author Response
Reviewer 3 (Point 1)
An interesting work with high scientific and clinical value. It deals with the effects of blood pressure on the condition of the brain in terms of volumetric parameters such as total white matter volume, total gray matter volume, focal changes volume and the volume of the right and left hippocampuses. It is based on a wide material and is analysed with well-selected statistical tools. The results are presented logically and clearly. The discussion is logical and communicative.
Response (Point 1)
We would like to thank the reviewer for taking the time to assess our manuscript. We have addressed all the points they raised below and throughout the manuscript.
Reviewer 3 (Point 2)
My only concern is the lack of information on possible cardiovascular risk factors. Neurological diseases are not the only ones responsible for the structural condition of the brain.
Pulse pressure alone does not explain the complexity of the effects of the heart and the large vessels state on the central nervous system; important are for example the condition of carotid arteries or the heart valves. Please refer to this suggestion in the methodology or write it in the limits of the study.
Response (Point 2)
We agree that besides neurological disorder, other factors can have an effect on brain structure including cardiovascular risk factors, and that of those, pulse pressure reflects only a fraction.
With this in mind, we have carefully controlled for the possible effects of some major cardiovascular, lifestyle, and socio-demographic risk factors in the regression analyses including age, sex, education, BMI, serum high-density lipoprotein cholesterol (HDL-C), total cholesterol (TC) level, diabetes mellitus diagnosed by a doctor, self-reported smoking status (never, previous, or current); level of physical activity and alcohol intake (page 3, line 132-135). However, we acknowledge that this may not have controlled for all possible effects of known or unknown risk factors. We have now added this limitation to the manuscript as follows:
“Although we have carefully controlled for the possible effects of some major cardiovascular, lifestyle, and socio-demographic risk factors in our analyses it is likely that this did not account fully for their influence or for the effects of other uncontrolled risk factors on brain structure (e.g., atherosclerosis, valvular heart disease). Consequently, future research should aim to include and contrast a greater variety of cardiovascular measures to achieve a more precise assessment of the effects of the heart and large vessels conditions on brain health. [page 17, line 388-394].
